# At What Point Are Long-Term (>5 Years) Survivors of APL Safe? A Study from the SEER Database

**DOI:** 10.3390/cancers15030575

**Published:** 2023-01-17

**Authors:** Xue-Jiao Yin, Rong Wang, Hong-Shi Shen, Jie Jin, Hong-Hu Zhu

**Affiliations:** 1Department of Hematology, The First Affiliated Hospital, Zhejiang University School of Medicine, Hangzhou 310003, China; 2Zhejiang Province Key Laboratory of Hematology Oncology Diagnosis and Treatment, Hangzhou 310003, China; 3Department of Hematology and Oncology, 904 Hospital of PLA, Wuxi 214000, China; 4Cancer Center, Zhejiang University, Hangzhou 310058, China; 5Zhejiang Laboratory for Systems & Precision Medicine, Zhejiang University Medical Center, Hangzhou 310000, China; 6Institute of Translational Medicine, Zhejiang University School of Medicine, Hangzhou 310029, China

**Keywords:** APL, standardized mortality ratios, SEER, causes of death, long-term survivors

## Abstract

**Simple Summary:**

The history of the treatment of APL has shown remarkable success. However, it is not clear whether it is necessary to monitor long-term toxicity in “cured” patients who survive for more than five years, which is critical to ensuring maximum survival in APL patients. To answer this question, we performed a comprehensive analysis of 1952 APL 5-year survivors and 5973 non-APL acute myeloid leukemia (AML) 5-year survivors from the Surveillance, Epidemiology, and End Results (SEER) database. We found that the overall mortality risk in >5-year survivors of APL was 1.41 times than that of the general population due to a significant increase in mortality from AML (SMR, 87.67) and other second malignant neoplasms (SMR, 1.56) only at 60–119 months. This result indicated long-term survivors of APL are safe after 10 years, which may help clinicians make future risk assessment, health guidance, and follow-up plans for APL long-term survivors.

**Abstract:**

Background: Acute promyelocytic leukemia (APL) is a highly curable cancer, but it is not clear whether it is also necessary to monitor long-term toxicity in “cured” patients who survive for more than five years, which is critical to ensuring maximum survival in APL patients. Methods: A total of 1952 APL 5-year survivors and 5973 non-APL acute myeloid leukemia (AML) 5-year survivors were included from the Surveillance, Epidemiology, and End Results (SEER) database. The standardized mortality ratio (SMR) was calculated to measure the risk of death. Cumulative mortality is calculated as the incidence of specific causes of death under competing risk events. Results: The SMR of all causes of death in >5-year survivors of APL was higher than that of the general population only at 60–119 months (SMR, 1.41). This was mainly because a significant increase in mortality from AML (SMR, 87.67) and second malignant neoplasms (SMNs) (SMR, 1.56) was found only at 60–119 months. However, there was no higher risk of death from non-cancer-related disease in >5-year survivors of APL than that of the general population (SMR, 0.89). The SMR of all-cause deaths in >5-year survivors of non-APL AML decreased year by year and was no higher than that of the general population until after 216 months. The cumulative incidence of AML-related death, SMN-related death, and non-cancer-related death was significantly lower in APL patients than in non-APL AML patients throughout the follow-up period. Conclusions: Compared with the general population, the risk of death of patients with APL was higher within 5 to 10 years but not higher over 10 years. Therefore, we believe that long-term survivors of APL are safe after 10 years.

## 1. Introduction

Acute promyelocytic leukemia (APL) is a very distinct acute myeloid leukemia (AML) subtype, and approximately 600–800 cases are newly diagnosed each year in the USA [1]. APL is characterized by a specific chromosomal balanced translocation t (15;17), along with the PML-RARA fusion gene [2,3]. APL was once one of the most life-threatening leukemia because of fatal bleeding risk due to hyperfibrinolysis, thrombocytopenia [4]. Since the advent of all-trans retinoic acid (ATRA) in the 1980s and arsenic trioxide (ATO) in the 1990s, outcomes of APL patients have improved dramatically. APL is now a highly curable disease with a complete remission rate of 90–95% [5,6], and the 5-year disease-free survival rate of over 80% [3,7,8]. In addition, with molecular monitoring of the minimal residual disease (MRD) assessment [9], physicians’ awareness of early mortality, and effective supportive care, the number of patients with APL who can achieve long-term survival of more than five years has become common.

Long-term survivors of cancer may die from delayed side effects due to treatment or cancer-related comorbidities, such as secondary tumors, cardiovascular disease, infections, diabetes, and psychiatric and social problems [10]. Some studies have found long-term side effects of APL therapy, including an increased risk of secondary tumors [11,12]. Therefore, it is important to know whether there is still a need to monitor “cured” APL patients with survival of more than five years, which is currently the standard to ensure maximum survival in APL patients. Temporal trends in causes of death and the risk of death are not well studied among 5-year or longer survivors at the population level.

Therefore, we performed a comprehensive analysis of the late effects of treatment or cancer itself in long-term APL survivors to guide follow-up screening programs. There were two principal objectives in this study: (1) to determine whether it is necessary to monitor long-term toxicity in APL “cured” patients who have survived for more than five years by comparing the mortality rates with the normal population, and (2) to explore the difference in long-term side effects between APL and non-APL AML, which had different therapeutic schedules.

## 2. Materials and Methods

### 2.1. Data Sources and Study Population Selection

The SEER database is currently the largest public database for cancer incidence statistics in the United States. It is used to collect the demographics, causes of death, and survival of cancer patients. All the data for this study come from the SEER-18 project registry. The registries of SEER-18 cover 18 regions in the US, accounting for approximately 28% of the population.

The inclusion criteria included (1) patients with APL and patients non-APL AML as their first tumors, (2) diagnosis time between 2000 and 2013, (3) survival of more than 5 years after diagnosis, (4) the last updated follow-up data at the end of 2018 (submitted in November 2020), which ensured that the included patients had at least 5 years of follow-up. APL histology codes included 9866, and non-APL AML histology codes included 9840, 9861, 9865, 9867, 9870–9874, 9891, 9895–9897, 9910, 9920, 9930, and 9931, using the third edition of the International Classification of Disease for Oncology (ICD-O-3) histology codes. Patients with the following criteria were excluded: (1) APL and non-APL AML were not primary cancers; (2) the cause of death was unknown. The present study conformed with the guidelines of the STROBE (Strengthening the Reporting of Observational Studies in Epidemiology) statement [13].

Data on sex, age, ethnicity, survival time, outcome, cause of death, and marital status were collected. Causes of death were divided into three categories: primary cancer-related death; secondary malignant neoplasm (SMN)-related death; and non-cancer disease-related death. The variable “cardiovascular disease mortality (CVM)” includes deaths caused by heart disease; atherosclerosis; hypertension without heart disease; aortic aneurysm and dissection; other diseases of the arteries, arterioles and capillaries; and cerebrovascular disease. The variable “infections” included the following: sepsis, pneumonia, and other infectious and parasitic diseases.

### 2.2. Statistical Analysis

The main parameter of the study is standardized mortality ratios (SMRs), which is the difference between the mortality rate observed in the cohort of the study and the expected mortality rate of the US general population within a similar age range, sex, and race/ethnicity during the same period. SMR is defined as “the general population mortality due to the additional risk of a specific disease”, and it provides a more direct and intuitive way to measure the impact of cancer on the risk of death. Since the study was only for 5-year survivors, the 5-year actual and expected mortality of the cohort studied was adjusted to 0%. SMR were calculated by SEER*Stat software, and the Poisson exact method was used to estimate the corresponding *p* value and 95% confidence interval (CI). A *p* value < 0.05 was considered the standard of statistical significance.

APL and non-APL AML have more than one endpoint event, and all other causes of death that are not related to a specific endpoint event are considered competing risk events. The competitive risk model could evaluate the cause of death of patients and could be used to calculate the incidence of endpoint events under the conditions of various end-point events competing with each other. R software (version: 3.2.5) was used to implement competitive risk models and were expressed in terms of cumulative mortality [14].

## 3. Results

### 3.1. Patient Characteristics

In total, 1952 APL >5-year survivors and 5973 non-APL AML >5-year survivors between 2000 and 2013 were included from the SEER database in this analysis (Table 1). The median age was 40.95 and 39.87 years for patients with APL and non-APL AML, respectively. The distribution of sex, race, and marital status of APL and non-APL AML survivors was similar. Married status (APL, 53.48%; non-APL AML, 49.36%) and non-Spanish-Hispanic-Latino ethnicity (APL, 77.66%; non-APL AML, 83.63%) account for a large proportion of these two groups.

In APL, 133 (6.81%) patients died at the end of follow-up. Of the deaths, 25 (18.8%) died of AML, 20 (15.03%) died of SMNs, and 88 (66.17%) died of noncancerous disease. In non-APL AML, 1030 (17.24%) patients died at the end of follow-up. Of the deaths, 411 (39.9%) died of AML, 192 (18.64%) died of SMNs, and 427 (41.46%) died of noncancerous diseases.

### 3.2. Comparison with the General Population

#### 3.2.1. Overall Mortality

The SMR of all-cause deaths in >5-year survivors of APL is higher than that of the general population only at 60–119 months. For example, the 60–119 month, 120–179 month, 180–215 month, and 216+ month SMRs of all-cause deaths in APL were 1.41 (95% C.I. 1.13–1.74); 1.22 (95% C.I. 0.87–1.66); 0.84 (0.31–1.83); and 0 (95% C.I. 0–10.73), respectively (Figure 1A). Only APL patients aged 15–39 (SMR, 2.35; 95% CI, 1.44–3.63) and 40–59 (SMR, 1.53; 95% CI, 1.15–2) had a higher overall risk of death than the general population.

The SMR of all-cause deaths in >5-year survivors of non-APL AML decreased year by year and was no higher than that of the general population after 216 months. For example, the 60–119 month, 120–179 month, 180–215 month, and 216+ month SMRs of all-cause deaths in non-APL AML were 3.89 (95% C.I. 3.61–4.18); 2.41 (95% C.I. 2.11–2.74); 1.71 (1.21–2.35); and 2.93 (95% C.I. 0.8–7.5), respectively (Figure 1B). The overall mortality risk of non-APL AML decreased with age. For example, APL diagnosed under 14 years of age (SMR, 11.71; 95% CI, 8.06–16.44) has a higher overall risk of death than APL diagnosed at 15–39 (SMR, 7.54; 95% CI, 6.34–8.89), 40–59 (SMR, 3.55; 95% CI, 3.2–3.92), and 60+ (SMR, 2.55; 95% CI, 2.32–2.79) years of age.

Among APL and non-APL AML survivors, female sex, and unmarried status were more likely to increase the overall mortality risk.

#### 3.2.2. Cause-Specific Mortality

Compared with the general population, a significant increase in mortality from AML (SMR, 87.67; 95% CI, 56.17–130.44) and SMNs (SMR, 1.56; 95% CI, 1.01–2.31) was found in >5-year survivors of APL only at months 60–119 (Table 2). For SMN causes of death, the SMR of liver and intrahepatic bile duct-related death was significantly elevated (SMR, 4.22; 95% CI, 1.37–9.85) (Figure 2A). Compared with male patients with APL, female patients with APL have a higher risk of dying from AML and SMNs. The SMR of AML-related death was higher among Spanish-Hispanic-Latino patients (SMR, 100.17; 95% CI, 36.76–218.02) than among Non-Spanish-Hispanic-Latino patients (SMR, 49.49; 95% CI, 29.79–77.28). There was no higher risk of death from non-cancer-related disease in >5-year survivors of APL than among the general population (SMR, 0.89; 95% CI, 0.69–1.13).

The risk of dying from AML (SMR, 292.93; 95% CI, 265.26–322.7) and SMNs (SMR, 2.84; 95% CI, 2.48–3.23) in >5-year survivors of non-APL AML was significantly higher than that of the general population (Table 3). The increased risk of dying from AML and SMNs decreased as the tracking time increased and was no higher than that of the general population after 216 months. For SMN death types, chronic myeloid leukemia-related death had the highest SMR (SMR, 124.90; 95% CI, 72.76–199.98). In addition, the SMRs of oral cavity and pharynx-related death (SMR, 3.58; 95% CI, 1.16–8.36), colon and rectum-related death (SMR, 1.99; 95% CI, 1.09–3.35), acute lymphocytic leukemia-related death (SMR, 16.56; 95% CI, 3.42–48.4), and chronic lymphocytic leukemia-related death (SMR, 7.51; 95% CI, 2.05–19.22) were also significantly elevated (Figure 2B). The SMR of non-cancer-related deaths in >5-year survivors of non-APL AML was higher than that of the general population only at 60–119 (SMR, 1.48; 95% CI, 1.28–1.69) and 120–179 months (SMR, 1.61; 95% CI, 1.33–1.93). For non-cancer-related death types, the SMRs of CVM (SMR, 1.29; 95% CI, 1.07–1.54) and infection-related death (SMR, 3.51; 95% CI, 2.6–4.63) were significantly elevated. The risk of dying from AML-related death, SMN-related deaths, and non-cancer-related deaths all decreased with age. Female sex, Spanish-Hispanic-Latino ethnicity, and unmarried status were more likely to have a higher SMR of AML-related death and non-cancer-related deaths.

### 3.3. Competing Risk Analyses

Competitive risk analysis was performed to evaluate the incidence of specific causes of death under the conditions of various endpoint events competing with each other among >5-year survivors of APL and non-APL AML (Figure 3). The cumulative incidence of AML-related death, SMN-related death, and non-cancer-related death was significantly lower in APL patients than in non-APL AML patients throughout the entire follow-up period.

The cumulative incidences of AML-related death, SMN-related death, and non-cancer-related death at 10 years were 1.59% (SD: 1.07 × 10^−5^), 0.79% (SD: 5.40 × 10^−6^), and 3.55% (SD: 2.41 × 10^−5^), respectively, among long-term APL survivors. The cumulative incidences of AML-related death, SMN-related death, and non-cancer-related death at 15 years were 1.92% (SD: 2.15 × 10^−5^), 1.70% (SD: 2.09 × 10^−5^), and 8.40% (SD: 9.53 × 10^−5^), respectively, among long-term APL survivors.

The cumulative incidences of AML-related death, SMN-related death, and non-cancer-related death at 10 years were 6.98% (SD: 1.33 × 10^−5^), 3.00% (SD: 5.97 × 10^−6^), and 5.57% (SD: 1.16 × 10^−5^), respectively, among long-term non-APL AML survivors. The cumulative incidences of AML-related death, SMN-related death, and non-cancer-related death at 15 years were 9.06% (SD: 2.16 × 10^−5^), 4.52% (SD: 1.21 × 10^−5^), and 11.14% (SD: 3.27 × 10^−5^), respectively, among long-term non-APL AML survivors.

## 4. Discussion

In this study, using the largest public cancer database, we found that the overall mortality risk in >5-year survivors of APL was 1.41 times that of the general population due to a significant increase in mortality from AML (SMR, 87.67) and SMNs (SMR, 1.56) only at 60–119 months after curative treatment. The overall mortality risk in >5-year survivors of non-APL AML decreased yearly and was no higher than that of the general population after 216 months. This result indicated long-term survivors of APL are safe after 10 years.

We found that compared with the general population, the risk of death of patients with APL was higher within 5 to 10 years but not higher over 10 years. Regardless of age and sex, the risk of death from APL is expected to decrease with the extension of follow-up time past 5 years. Many previous articles have reported that secondary cancers and recurrences after APL treatment occur within the latency period of several months to several years, which is closely related to the treatment strategy for APL. For example, a study from the Albert Einstein Cancer Center showed that most APL patients have the greatest risk of recurrence within 3 years after completion of treatment [15]. Ng, A.K. et al., reported that children and adults who receive a chemotherapy regimen that includes topoisomerase II inhibitors are at increased risk of treatment-related AML (t-AML), usually within 3 years of exposure [16]. S. Zompi et al., reported that the median incubation period for APL from complete remission to the diagnosis of t-AML was 34 (25–40) months [17].

In this study, we found that >5-year survivors of non-APL AML had a significantly higher risk of dying from heart-related causes (SMR, 1.29; 95% CI, 1.07–1.54) than the general population, while APL did not (SMR, 1.00; 95% CI, 0.67–1.42). This phenomenon could be attributed to low-intensity chemotherapy regimens and intensive chemotherapy regimens administered to APL patients and non-APL AML patients [18]. Compared with the standard chemotherapy regimens administered to non-APL AML patients, the dose of anthracyclines contained in APL treatment is greatly reduced [18,19]. Anthracyclines have been essential for AML treatment for decades [20,21], contributing significantly to improved survival, especially in patients with APL, where anthracyclines have proven to be the gold standard of chemotherapy during 2000–2013 in the USA [22]. Unfortunately, patients receiving high cumulative doses of anthracyclines have a higher risk of congestive heart failure and mortality, and the risk of heart failure increases exponentially with the dose of anthracyclines [23]. Previous studies have shown that receiving a higher dose of anthracyclines in the standard intensive chemotherapy regimen for non-APL AML is associated with higher remission rates and lower overall mortality [24], but patients are at greater risk of developing cardiac injury. Another important reason was that ATRA does not cause specific cardiotoxicity [25]. However, anthracyclines in combination with other cardiotoxic drugs, such as amsacrine and cyclophosphamide, were used in the standard regimen for the treatment of non-APL AML, which will greatly increase the risk of heart disease and mortality [26,27]. A higher risk of cardiotoxicity with anthracycline has been reported when amsacrine is used after anthracycline therapy among AML patients [28].

Our study found that patients with APL had a significantly lower risk of dying from AML (SMR, 56.33; 95% CI, 36.45–83.15) and SMN (SMR, 1.48; 95% CI, 1.05–2.03) than patients with non-APL AML (SMR for AML, 292.93; 95% CI, 265.26–322.7; SMR for SMN, 2.84; 95% CI, 2.48–3.23). These findings are expected, and the reasons may be as follows: (1) patients with non-APL AML will receive higher doses of anthracyclines, topoisomerase II targeted drugs or than APL patients, which may cause resistant clone selection and DNA damage that may be closely related to the recurrence and the development of secondary AML [29]; (2) compared with non-APL AML, less APL patients received HSCT which included high-dose conditioning or total body irradiation.

For SMN causes of death in >5-year survivors of APL, the SMR of liver and intrahepatic bile duct-related death was significantly elevated (SMR, 4.22; 95% CI, 1.37–9.85). A long-term follow-up study showed that the incidence of liver dysfunction and steatosis in 217 newly diagnosed APL patients was significantly higher than that of healthy control patients [30]. Another study also showed that hepatotoxicity, especially complications of increased liver enzymes (mainly AST and ALT and less commonly alkaline phosphatase and bilirubin), may occur in up to 60% of cases receiving APL treatment [31]. Luana et al. reported that the relative risk of secondary liver tumors in patients with APL was significantly higher than that in patients with non-APL AML or other tumors [11]. This higher incidence of secondary liver cancer may potentially result in an increase in the mortality rate from secondary liver cancer. For SMN causes of death in >5-year survivors of non-APL AML, the SMR of oral cavity and pharynx-related cancer death (SMR, 3.58; 95% CI, 1.16–8.36) and colon and rectum-related cancer death (SMR, 1.99; 95% CI, 1.09–3.35) was significantly elevated. A previous study reported that compared with the general population, AML patients in different age groups were at significantly higher risk of developing cancers of the oral cavity, pharynx, and gastrointestinal system [32]. The follow-up screening plans of long-term survivors of non-APL AML and APL should focus on the early screening of these cancer types.

The results of this study may indicate that for APL patients, arsenic trioxide (ATO) + ATRA without combined chemotherapy may be a more appropriate choice. Lo-Coco et al. demonstrated that APL patients can be cured without conventional chemotherapy [33]. Long-term data from Y. Abaza’s team confirmed the efficacy, safety, and durability of ATRA plus ATO chemotherapy-free strategies to provide durable disease-free survival in standard and high-risk patients [34]. Furthermore, a follow-up study by Zhu et al. confirmed that ATRA/ATO combination therapy could achieve a 7-year survival rate of more than 90%, with no significant difference in prognosis between high-risk and non-high-risk patients [35]. In a 12-year long-term follow-up study of 265 newly diagnosed APL patients treated with ATRA and ATO, no chronic adverse reactions, such as cardiovascular events, chronic renal insufficiency, or diabetic neurological dysfunction, were observed, and only one patient developed breast cancer, which was not associated with ATO treatment [36]. From these data above, ATRA plus ATO without chemotherapy may have a lower risk of secondary cancer and non-cancer-related death, which need to be confirmed by studies with larger samples and longer follow-up.

There were some limitations in this study. The main limitation is that the SEER database does not contain the patient’s treatment information, so it is difficult to know the specific chemotherapy regimen for each patient, such as drug selection and dose. Second, there are no data on minimal residual disease (MRD) in the SEER database, which is closely related to disease progression. Third, there is a lack of records of patients’ specific symptoms, karyotypes, and personal habits, including alcohol consumption, smoking, and obesity, which are associated with the cause of death and mortality of APL.

## 5. Conclusions

In summary, compared with the general population, the risk of death of patients with APL was higher within 5 to 10 years but not higher beyond 10 years. Therefore, we believe that long-term survivors of APL are safe after 10 years.

## Figures and Tables

**Figure 1 cancers-15-00575-f001:**
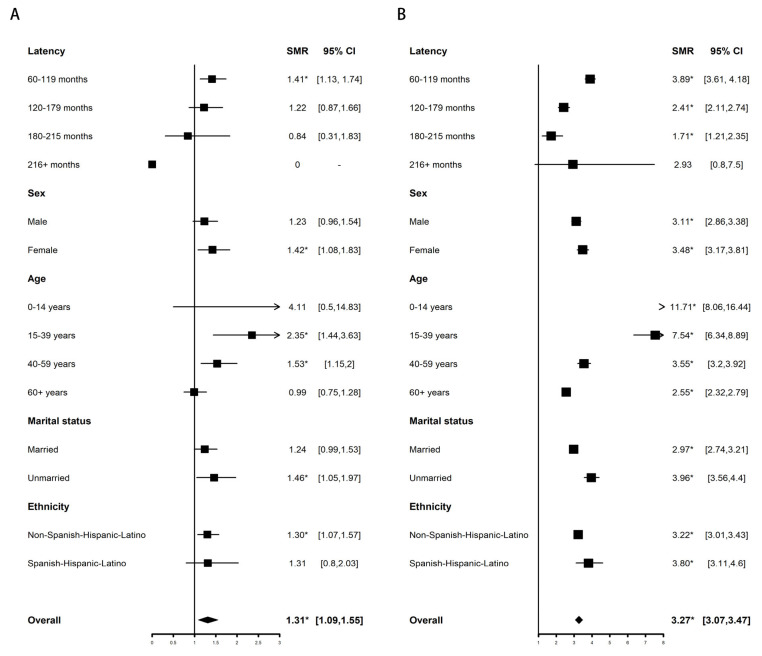
The SMRs of all-cause deaths among >5-year survivors of APL (**A**) and non-APL AML (**B**) compared with the general population stratified by demographics and clinical characteristics. The solid vertical lines represent the SMR of 1.0. The horizontal lines represent 95% confidence intervals for the SMRs. The box represents the result of SMR. The diamond at the bottom shows the summary SMR of all the combined individual studies. We used the Poisson exact method for SMR to compute the 95% CI and corresponding *p* values, and the 95% CIs not including 1 with *p* < 0.05 were defined as significant differences for SMR. * *p* < 0.05.

**Figure 2 cancers-15-00575-f002:**
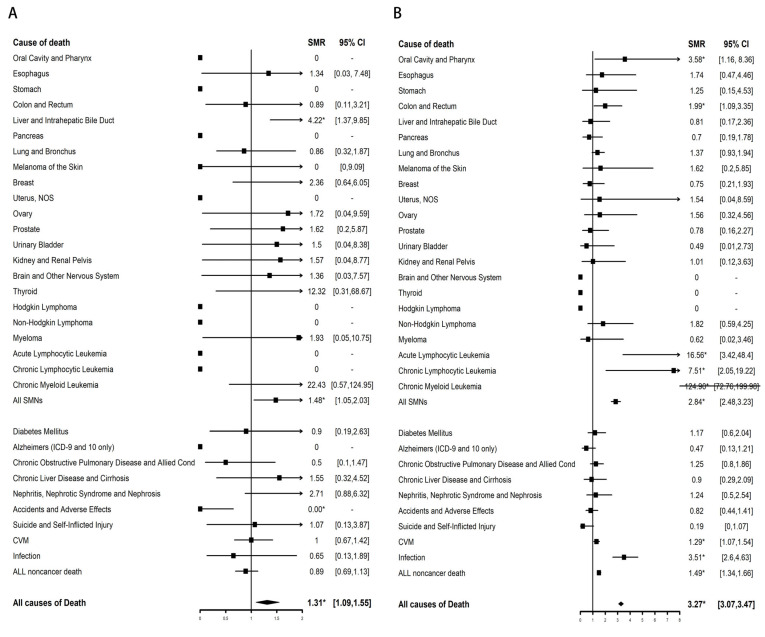
SMRs of site-specific death among >5-year survivors of APL (**A**) and non-APL AML (**B**) compared with the general population. The solid vertical lines represent the SMR of 1.0. The horizontal lines represent 95% confidence intervals for the SMRs. The box represents the result of SMR. The diamond at the bottom shows the summary SMR of all the combined individual studies. We used the Poisson exact method for SMR to compute the 95% CI and corresponding *p* values, and the 95% CIs not including 1 with *p* < 0.05 were defined as significant differences for SMR. * *p* < 0.05.

**Figure 3 cancers-15-00575-f003:**
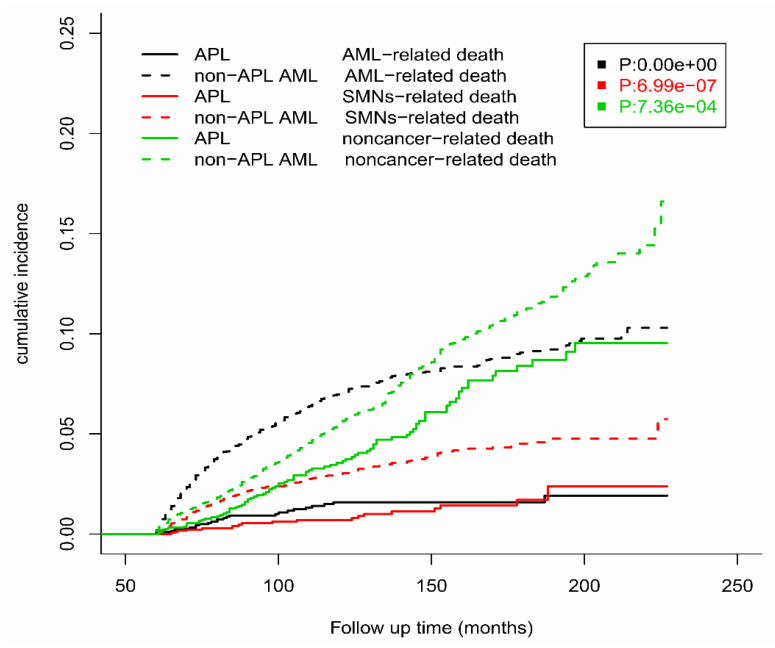
The cumulative incidences of AML-related death, SMN-related death, and non-cancer-related death among >5-year survivors of APL and non-APL AML.

**Table 1 cancers-15-00575-t001:** Demographics and clinical characteristics of the patients.

Characteristic	APL	Non-APL AML
Overall	1952	5973
Age at diagnosis, y		
0–14	128 (6.56%)	991 (16.59%)
15–39	778 (39.86%)	1621 (27.14%)
40–59	742 (38.01%)	2182 (36.53%)
60+	304 (15.57%)	1179 (19.74%)
Sex		
Male	968 (49.59%)	3057 (51.18%)
Female	984 (50.41%)	2916 (48.82%)
Ethnicity		
Non-Spanish-Hispanic-Latino	1516 (77.66%)	4995 (83.63%)
Spanish-Hispanic-Latino	436 (22.34%)	978 (16.37%)
Marital status		
Married	1044 (53.48%)	2948 (49.36%)
Unmarried	844 (43.24%)	2835 (47.46%)
Unknown	64 (3.28%)	190 (3.18%)
Alive	1819 (93.19%)	4943 (82.76%)
Dead	133 (6.81%)	1030 (17.24%)
Died from AML ^a^	25 (18.8%)	411 (39.9%)
Died from SMNs ^a^	20 (15.03%)	192 (18.64%)
Died from noncancer disease ^a^	88 (66.17%)	427 (41.46%)
Cardiovascular disease ^b^	30 (34.09%)	120 (28.1%)
Infection ^b^	3 (3.41%)	51 (11.95%)
Other non-cancer disease ^b^	55 (62.5%)	256 (59.95%)

^a^: Percentage of total deaths. ^b^: Percentage of non-cancer related deaths.

**Table 2 cancers-15-00575-t002:** Standardized mortality ratios (SMRs) for 5-year survivors of APL according to baseline characteristics.

Characteristic	AML	SMN	All Non-Cancer Disease	CVM	Infection
	SMR (95% CI)	SMR (95% CI)	SMR (95% CI)	SMR (95% CI)	SMR (95% CI)
Overall	56.33 * (36.45–83.15)	1.48 * (1.05–2.03)	0.89 (0.69–1.13)	1 (0.67–1.42)	0.65 (0.13–1.89)
Age at diagnosis, y					
0–14	419.59 * (10.62–2337.79)	0 (0–134.5)	2.19 (0.06–12.22)	0 (0–167.68)	0 (0–373.02)
15–39	122.77 * (33.45–314.34)	4.11 * (1.51–8.94)	1.43 (0.69–2.63)	2.6 (0.71–6.65)	0 (0–9.69)
40–59	73.02 * (37.73–127.56)	1.47 (0.84–2.38)	0.96 (0.61–1.44)	1.05 (0.5–1.92)	0.6 (0.02–3.36)
60+	32.71 * (14.12–64.46)	1.2 (0.69–1.96)	0.75 (0.52–1.06)	0.84 (0.48–1.37)	0.77 (0.09–2.8)
Sex					
Male	54.06 * (30.26–89.16)	0.94 (0.51–1.58)	0.98 (0.71–1.32)	1.14 (0.7–1.74)	0.37 (0.01–2.08)
Female	60.11 * (28.82–110.54)	2.23 * (1.43–3.32)	0.76 (0.48–1.13)	0.77 (0.35–1.47)	1.03 (0.12–3.71)
Ethnicity					
Non-Spanish-Hispanic-Latino	49.49 * (29.79–77.28)	1.44 (0.98–2.03)	0.93 (0.71–1.2)	1.05 (0.69–1.53)	0.76 (0.16–2.21)
Spanish-Hispanic-Latino	100.17 * (36.76–218.02)	1.77 (0.65–3.85)	0.68 (0.3–1.35)	0.68 (0.14–1.99)	0 (0–5.53)
Latency periods, m					
60–119.	87.67 * (56.17–130.44)	1.56 * (1.01–2.31)	0.8 (0.56–1.11)	0.83 (0.47–1.38)	0.71 (0.09–2.56)
120–179	0 (0–26.35)	1.38 (0.69–2.46)	1.19 (0.8–1.71)	1.53 (0.85–2.52)	0.68 (0.02–3.77)
180–215	34.78 (0.88–193.77)	1.24 (0.15–4.49)	0.37 (0.04–1.33)	0 (0–1.67)	0 (0–11.58)
216+	0 (0–2777.23)	0 (0–46.19)	0 (0–14.15)	0 (0–36.14)	0 (0–235.36)
Marital status					
Married	59.89 * (36.06–93.52)	1.26 (0.8–1.9)	0.84 (0.61–1.13)	1.05 (0.66–1.59)	0.63 (0.08–2.29)
Unmarried	53.52 * (19.64–116.5)	2.12 * (1.16–3.55)	1 (0.63–1.52)	0.85 (0.34–1.74)	0 (0–2.75)

* *p* < 0.05.

**Table 3 cancers-15-00575-t003:** Standardized mortality ratios (SMRs) for 5-year survivors of non-APL AML according to baseline characteristics.

Characteristic	AML	SMN	All Non-Cancer Disease	CVM	Infection
	SMR (95% CI)	SMR (95% CI)	SMR (95% CI)	SMR (95% CI)	SMR (95% CI)
Overall	292.93 * (265.26–322.7)	2.84 * (2.48–3.23)	1.49 * (1.34–1.66)	1.29 * (1.07–1.54)	3.51 * (2.6–4.63)
Age at diagnosis, y					
0–14	1159.10 * (675.22–1855.83)	40.20 * (16.16–82.84)	3.44 * (1.57–6.53)	0 (0–29.28)	32.86 * (3.98–118.68)
15–39	938.31 * (725.69–1193.75)	9.90 * (6.73–14.05)	2.72 * (1.96–3.68)	2.63 * (1.2–5)	8.22 * (3.31–16.94)
40–59	290.97 * (246.7–340.91)	2.32 * (1.84–2.88)	1.92 * (1.61–2.27)	1.31 (0.93–1.8)	5.22 * (3.41–7.65)
60+	220.58 * (189.02–255.9)	2.59 * (2.13–3.12)	1.1 (0.93–1.29)	1.2 (0.94–1.51)	1.80 * (1–2.96)
Sex					
Male	249.29 * (217.01–285.02)	2.93 * (2.45–3.46)	1.43 * (1.23–1.64)	1.2 (0.93–1.53)	3.09 * (2–4.57)
Female	362.14 * (313.22–416.55)	2.72 * (2.2–3.32)	1.58 * (1.34–1.85)	1.41 * (1.05–1.84)	4.05 * (2.62–5.98)
Ethnicity					
Non-Spanish-Hispanic-Latino	285.96 * (257.33–316.9)	2.87 * (2.5–3.28)	1.46 * (1.3–1.63)	1.23 * (1.01–1.49)	3.44 * (2.51–4.61)
Spanish-Hispanic-Latino	362.91 * (265.69–484.07)	2.51 * (1.49–3.97)	1.84 * (1.29–2.53)	1.90 * (1.04–3.19)	4.23 * (1.37–9.87)
Latency periods, m					
60–119	395.35 * (354.9–439.15)	3.22 * (2.75–3.75)	1.48 * (1.28–1.69)	1.30 * (1.02–1.63)	4.10 * (2.87–5.68)
120–179	123.33 * (92.11–161.72)	2.18 * (1.63–2.85)	1.61 * (1.33–1.93)	1.37 (0.98–1.87)	2.94 * (1.56–5.03)
180–215	108.31 * (51.94–199.19)	2.11 * (1.05–3.77)	1.01 (0.59–1.62)	0.89 (0.33–1.94)	1.01 (0.03–5.6)
216+	0 (0–676.46)	3.29 (0.08–18.35)	2.86 (0.59–8.36)	0 (0–8.7)	0 (0–61.53)
Marital status					
Married	259.25 * (228.69–292.76)	2.43 * (2.05–2.87)	1.32 * (1.14–1.51)	1.2 (0.94–1.5)	3.33 * (2.28–4.71)
Unmarried	381.37 * (319.54–451.69)	3.96 * (3.15–4.91)	1.88 * (1.57–2.23)	1.56 * (1.12–2.11)	4.12 * (2.4–6.6)

* *p* < 0.05.

## Data Availability

The data produced and analyzed in the current study are all available in the SEER database (https://seer.cancer.gov/seerstat/, data accessed on 29 October 2021), which is freely available to the public.

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
