# Peer review of "At What Point Are Long-Term (>5 Years) Survivors of APL Safe? A Study from the SEER Database"

_cancers, 2023, doi:10.3390/cancers15030575_

Round 1
Reviewer 1 Report
The authors addressed the long-term survival of acute promyelocytic patients (APL), compared to the non-APL AML cohort and the general population of USA SEER database. I think the topic is clinically relevant for physicians that treat and follow APL patients. This study evaluated a large APL population.
The authors used public data from SEER USA database to evaluate
standardized mortality ratio of APL patients compared to general population and non-APL AML. They found that APL patients had a lower mortality due to secondary malignancy and cardiotoxicity and that the risk of death of APL patients after 10 years is similar to the general population. Those data are clinically relevant for the follow-up and counseling of APL patients.
The manuscript is well written and the references are appropriate for the discussion of APL in Europe and United States. It would be interesting to discuss APL long term outcomes in other parts of the world, such as Latin America or other developing countries.
Author Response
Date: Jan. 4, 2022
Dear Dr. Liu,
On behalf of all the contributing authors, I would like to express our sincere appreciation for your letter and reviewers’ constructive comments concerning our manuscript entitled " At what point are long-term (>5 years) survivors of APL safe? - A study from the SEER database " (Manuscript ID: cancers-2085755). These comments are all valuable and helpful for improving our manuscript. We have made modifications according to the comments. In this revised version, changes to our manuscript are highlighted using red text.
We hope that our revised manuscript will be considered for publication in Cancers. Thank you very much for your help!
Sincerely yours,
Hong-Hu Zhu.
Professor
Department of Hematology, The First Affiliated Hospital, College of Medicine, Zhejiang University, #79 Qingchun Road, Hangzhou 310003, China.
Tel: 0086-571-87236717
Fax: 0086-571-87236717
E-mail: zhuhhdoc@163.com
Point-by-point responses are listed below.
Reviewer: 1
Comments to the Author
- The authors addressed the long-term survival of acute promyelocytic patients (APL), compared to the non-APL AML cohort and the general population of USA SEER database. I think the topic is clinically relevant for physicians that treat and follow APL patients. This study evaluated a large APL population.
The authors used public data from SEER USA database to evaluate
standardized mortality ratio of APL patients compared to general population and non-APL AML. They found that APL patients had a lower mortality due to secondary malignancy and cardiotoxicity and that the risk of death of APL patients after 10 years is similar to the general population. Those data are clinically relevant for the follow-up and counseling of APL patients.
The manuscript is well written and the references are appropriate for the discussion of APL in Europe and United States. It would be interesting to discuss APL long term outcomes in other parts of the world, such as Latin America or other developing countries.
Answer: Thank you for the suggestion. We have modified the corresponding text in the revised manuscript according to your comment. In the present study, we collected information on APL patients from the registries of the Surveillance, Epidemiology, and End Results database (SEER) program of the US National Cancer Institute. The SEER database is the primary source of cancer incidence statistics in the United States, which gathering and reporting demographics, morphology, primary tumor site, treatment information, and survival data of patients with tumors. It covers up to 28% of the population in the US, including 67% Pacific/Hawaiian Islander, 50% Asian, 44% Alaskan/American Indian Native, 38% Hispanic, and 26% black. Although this analysis was based purely on the US population, the prevalence of APL long term outcomes in other parts of the world may have a certain degree of similarity. Special thanks to you for your comments.
We hope that the revised manuscript will meet your approval. Thank you!

Reviewer 2 Report
The authors compared the incidence and kind of complications, mortality , second cancers after the treatment for AML and APL-AML, based on data found in SEER-18. The negative side of this work is the complete lack of information on treatment, including stem cell transplantation, cranial radiotherapy, kind of chemotherapy (that has changed over the years!).Have all APL-AML patients received ATRA+/-ATO+/- chemotherapy? Different protocols were used in children and adults which also affects the type of complications. The kind of treatment has a decisive role on the future state of health. The lack of the data on the method of treatment may cause the difficulties in distinguishing the causes of death, second cancers and different organs function.
The lack of this data means that in the discussion the authors refer mainly the other information from the literature.
In the tables/figures please add: age at the time of diagnosis
Author Response
Date: Jan. 4, 2022
Dear Dr. Liu,
On behalf of all the contributing authors, I would like to express our sincere appreciation for your letter and reviewers’ constructive comments concerning our manuscript entitled " At what point are long-term (>5 years) survivors of APL safe? - A study from the SEER database " (Manuscript ID: cancers-2085755). These comments are all valuable and helpful for improving our manuscript. We have made modifications according to the comments. In this revised version, changes to our manuscript are highlighted using red text.
We hope that our revised manuscript will be considered for publication in Cancers. Thank you very much for your help!
Sincerely yours,
Hong-Hu Zhu.
Professor
Department of Hematology, The First Affiliated Hospital, College of Medicine, Zhejiang University, #79 Qingchun Road, Hangzhou 310003, China.
Tel: 0086-571-87236717
Fax: 0086-571-87236717
E-mail: zhuhhdoc@163.com
Point-by-point responses are listed below.
Reviewer: 2
Comments to the Author
1)The authors compared the incidence and kind of complications, mortality , second cancers after the treatment for AML and APL-AML, based on data found in SEER-18. The negative side of this work is the complete lack of information on treatment, including stem cell transplantation, cranial radiotherapy, kind of chemotherapy (that has changed over the years!).Have all APL-AML patients received ATRA+/-ATO+/- chemotherapy? Different protocols were used in children and adults which also affects the type of complications. The kind of treatment has a decisive role on the future state of health. The lack of the data on the method of treatment may cause the difficulties in distinguishing the causes of death, second cancers and different organs function.
The lack of this data means that in the discussion the authors refer mainly the other information from the literature.
Answer: Thank you for the suggestion. It is true that the kind of treatment has a decisive role on the future state of health and should be examined and evaluated. However, some limitations, based on the information available in the SEER database, should be taken into account when interpreting our findings. The main limitation is that the SEER database does not contain the patient's treatment information, including stem cell transplantation, cranial radiotherapy and kind of chemotherapy, so it is difficult to know the specific chemotherapy regimen for each patient, such as drug selection and dose. Nevertheless, retinoic acid combined with arsenic and low-intensity chemotherapy regimens for APL have proven to be the gold standard treatment during 2000-2013 in the USA. The limitations have been described in detail in the discussion section.
2)In the tables/figures please add: age at the time of diagnosis
Answer: Thank you for the suggestion. We have modified the corresponding text in the revised manuscript according to your comment.
We hope that the revised manuscript will meet your approval. Thank you!

Reviewer 3 Report
This manuscript, written by Dr. Yin, original report, with the title of "At what point are long-term (>5 years) survivors of APL safe? - A study from the SEER database" analyzed data obtained from the Surveillance, Epidemiology, and End Results (SEER) database. It focused on acute promyelocytic leukemia (APL) and aimed to elucidate if after more than 5 years the mortality was different than the general population.
The manuscript is well written, and it is easy to understand. Nevertheless, it is using the SEER database, so it is not their own data.
1) Did you compare APL vs. healthy population or vs. non-APL acute myeloid leukemia?
2) There are many abbreviations in the text, I would recommend reducing them or write an abbreviation list at the end of the discussion.
3) It may help the readers if it is explained how to interpret the "forest plots" of Figures 1 and 2.
4) Could you please add the reference for the R programs?
5) The risk of death was not higher after 10 years, in comparison to non-APL AML or the general population? After 10 years, did they have higher risk of secondary diseases or secondary neoplasia?
Author Response
Date: Jan. 4, 2022
Dear Dr. Liu,
On behalf of all the contributing authors, I would like to express our sincere appreciation for your letter and reviewers’ constructive comments concerning our manuscript entitled " At what point are long-term (>5 years) survivors of APL safe? - A study from the SEER database " (Manuscript ID: cancers-2085755). These comments are all valuable and helpful for improving our manuscript. We have made modifications according to the comments. In this revised version, changes to our manuscript are highlighted using red text.
We hope that our revised manuscript will be considered for publication in Cancers. Thank you very much for your help!
Sincerely yours,
Hong-Hu Zhu.
Professor
Department of Hematology, The First Affiliated Hospital, College of Medicine, Zhejiang University, #79 Qingchun Road, Hangzhou 310003, China.
Tel: 0086-571-87236717
Fax: 0086-571-87236717
E-mail: zhuhhdoc@163.com
Point-by-point responses are listed below.
Reviewer: 3
Comments to the Author
This manuscript, written by Dr. Yin, original report, with the title of "At what point are long-term (>5 years) survivors of APL safe? - A study from the SEER database" analyzed data obtained from the Surveillance, Epidemiology, and End Results (SEER) database. It focused on acute promyelocytic leukemia (APL) and aimed to elucidate if after more than 5 years the mortality was different than the general population.
The manuscript is well written, and it is easy to understand. Nevertheless, it is using the SEER database, so it is not their own data.
1) Did you compare APL vs. healthy population or vs. non-APL acute myeloid leukemia?
Answer: Thank you for the suggestion. Cause-specific mortality was assessed between APL and healthy people, and between non-APL AML and the general population via SMR. SMRs provide the relative risk of death for patients with cancer compared to the US general residents, which was characterized adjusted by age, race, and sex to the US general population over the same time. SMR is defined as “the general population mortality due to the additional risk of a specific disease”, and it provides a more direct and intuitive way to measure the impact of cancer on the risk of death.
2) There are many abbreviations in the text, I would recommend reducing them or write an abbreviation list at the end of the discussion.
Answer: Thank you for the suggestion. We have modified the corresponding text in the revised manuscript according to your comment.
3) It may help the readers if it is explained how to interpret the "forest plots" of Figures 1 and 2.
Answer: Thank you for the suggestion. We have modified the corresponding text in the revised manuscript according to your comment. Figure 1. The SMRs of all-cause deaths among >5-year survivors of APL (A) and non-APL AML (B) compared with the general population stratified by demographics and clinical characteristics. The solid vertical lines represent the SMR of 1.0. The horizontal lines represent 95% confidence intervals for the SMRs. The box represents the result of SMR. The diamond at the bottom shows the summary SMR of all the combined individual studies. We used the Poisson exact method for SMR to compute the 95% CI and corresponding P values, and the 95% CIs not including 1 with P < 0.05 were defined as significant differences for SMR. Figure 2. SMRs of site-specific death among > 5-year survivors of APL (A) and non-APL AML (B) compared with the general population. The solid vertical lines represent the SMR of 1.0. The horizontal lines represent 95% confidence intervals for the SMRs. The box represents the result of SMR. The diamond at the bottom shows the summary SMR of all the combined individual studies. We used the Poisson exact method for SMR to compute the 95% CI and corresponding P values, and the 95% CIs not including 1 with P < 0.05 were defined as significant differences for SMR.
4) Could you please add the reference for the R programs?
Answer: Thank you for the suggestion. We have modified the corresponding text in the revised manuscript according to your comment.
5) The risk of death was not higher after 10 years, in comparison to non-APL AML or the general population? After 10 years, did they have higher risk of secondary diseases or secondary neoplasia?
Answer: Thank you for the suggestion. Compared with the general population, the risk of all-cause deaths of patients with APL was not higher after 10 years. For example, the 60-119 month, 120-179 month, 180-215 month, and 216+ month SMRs of all-cause deaths in APL were 1.41 (95% C.I. 1.13-1.74); 1.22 (95% C.I. 0.87-1.66); 0.84 (0.31-1.83); and 0 (95% C.I. 0-10.73), respectively (Figure 1A). There was no higher risk of death from non-cancer-related disease and second malignant neoplasms in APL than that of the general population after 10 years. For example, the 60-119 month, 120-179 month, 180-215 month, and 216+ month SMRs of non-cancer-related disease in APL were 0.80 (95% C.I. 0.56-1.11); 1.19 (95% C.I. 0.8-1.71); 0.37 (0.04-1.33); and 0 (95% C.I. 0-14.15), respectively (Table 2). The 60-119 month, 120-179 month, 180-215 month, and 216+ month SMRs of second malignant neoplasms in APL were 1.56 (95% C.I. 1.01-2.31); 1.38 (95% C.I. 0.69-2.46); 1.24 (0.15-4.49); and 0 (95% C.I. 0-46.19), respectively (Table 2). We used the Poisson exact method for SMR to compute the 95% CI and corresponding P values, and the 95% CIs not including 1 with P < 0.05 were defined as significant differences for SMR.
We hope that the revised manuscript will meet your approval. Thank you!
